# Logical Languages Accepted by Transformer Encoders with Hard Attention

**Pablo Barceló**
IMC, PUC Chile & IMFD Chile & CENIA
Santiago, Chile
pbarcelo@uc.cl

**Alexander Kozachinskiy**
IMFD and CENIA
Santiago, Chile.
alexander.kozachinskyi@cenia.cl

**Anthony Widjaja Lin**
University of Kaiserslautern-Landau
& Max-Planck Institute for Software Systems
Kaiserslautern, Germany
awlin@mpi-sws.org

**Vladimir Podolski**
Tufts University
Medford, USA
vladimir.podolskii@tufts.edu

## Abstract

We contribute to the study of formal languages that can be recognized by transformer encoders. We focus on two self-attention mechanisms: (1) UHAT (Unique Hard Attention Transformers) and (2) AHAT (Average Hard Attention Transformers). UHAT encoders are known to recognize only languages inside the circuit complexity class $\mathsf{AC}^0$, i.e., accepted by a family of poly-sized and depth-bounded boolean circuits with unbounded fan-ins. On the other hand, AHAT encoders can recognize languages outside $\mathsf{AC}^0$), but their expressive power still lies within the bigger circuit complexity class $\mathsf{TC}^0$, i.e., $\mathsf{AC}^0$-circuits extended by majority gates. We first show a negative result that there is an $\mathsf{AC}^0$-language that cannot be recognized by an UHAT encoder. On the positive side, we show that UHAT encoders can recognize a rich fragment of $\mathsf{AC}^0$-languages, namely, all languages definable in first-order logic with arbitrary unary numerical predicates. This logic, includes, for example, all regular languages from $\mathsf{AC}^0$. We then show that AHAT encoders can recognize all languages of our logic even when we enrich it with counting terms. Using these results, we obtain a characterization of which counting properties are expressible by UHAT and AHAT, in relation to regular languages.

## 1 Introduction

Transformers have revolutionized natural language processing by facilitating the efficient and effective modeling of intricate contextual relationships within text (Vaswani et al., 2017). This remarkable capability has sparked numerous investigations into the potential boundaries of transformers' power (Hahn, 2020; Yao et al., 2021; Pérez et al., 2021; Weiss et al., 2021; Hao et al., 2022; Chiang & Cholak, 2022; Bhattamishra et al., 2020; Chiang et al., 2023; Merrill et al., 2022; Merrill & Sabharwal, 2023; Strobl, 2023). One natural method for addressing this question is to explore the classes of formal languages that these architectures can recognize. This approach provides an insight into their strengths and limitations. The response to this question naturally relies on the specific features allowed within transformer encoders. These encompass the interplay between encoders and decoders, the kind of functions used for positional encodings and attention mechanisms, and considerations of fixed or unbounded precision, among other factors.

While the capacity of transformers that incorporate both encoders and decoders to recognize languages is well understood today (indeed, such architectures are Turing-complete and can thus recognize any computable language (Pérez et al., 2021)), a precise characterization of the languages accepted by transformer encoders is lacking. *Unique Hard Attention Transformers (UHAT)* are a class of transformer encoders that has been a subject of many recent papers. As was shown by Hao et al. (2022), UHATs recognize only languages in $\mathsf{AC}^0$, i.e., recognized by families of Boolean circuits of unbounded fan-in that have constant depth and polynomial size. Intuitively, this means that

UHATs are weak at "counting" (more precisely, reasoning about the number of occurrences of various letters in the input word). For example, consider the following languages: *majority* and *parity*. The first one corresponds to the set of words over alphabet $\{a, b\}$ for which the majority of positions are labeled by $a$, while the second checks if the number of positions labeled $a$ is even. That these languages are not in $AC^0$ follows from a groundbreaking result in circuit complexity theory (Furst et al., 1981; Ajtai, 1983)). Hence, they are neither accepted by UHATs. However, which fragment of the $AC^0$ languages can actually be recognized by UHATs remains an unresolved question.

We start by showing that not all $AC^0$ languages can be accepted by UHATs. This is obtained by combining results of Ajtai (1983) and Hahn (2020). Based on the previous observation, we focus on identifying a rich fragment of $AC^0$ that can in fact be embedded into the class of UHATs. To achieve this, we use the characterization of $AC^0$ as the class of languages expressible in FO(All), the extension of first-order logic (FO) with all numerical predicates defined in relation to the linear order of a word (Immerman, 1999). We show that UHATs recognize all languages definable in FO(Mon), the restriction of FO(All) with *unary* numerical predicates only (Barrington et al., 2005). The logic FO(Mon) is highly expressive. Unlike FO, it can express non-regular languages like $\{a^n b^n \mid n > 0\}$. Remarkably, it contains all *regular languages* within $AC^0$, which includes examples like $(aa)^*$ — a language not definable in FO. Additionally, our result subsumes the result of Yao et al. (2021), where it is shown that *Dyck languages* of bounded nested depth can be recognized by UHATs. These languages are regular and belong to $AC^0$, hence they are expressible in FO(Mon).

To establish the result that UHATs recognize all languages definable in FO(Mon), we take a slightly circuitous route: rather than directly formulating FO(Mon) sentences as UHATs, we show that each formula in LTL(Mon), the extension of *linear temporal logic* (LTL) (Clarke et al., 2018) with arbitrary unary numerical predicates, can be equivalently represented as an UHAT. The proof for FO(Mon) then derives from Kamp's seminal theorem (Kamp, 1968), which establishes the equivalence between languages definable in FO and LTL. The advantage of dealing with LTL, in contrast to FO, lies in the fact that all LTL formulas are unary in nature, i.e., they are interpreted as sets of positions on a word, unlike FO formulas which possess arbitrary arity. This property aligns well with the expressive capabilities of UHATs, facilitating a proof through structural induction.

While the fact that UHAT is in $AC^0$ implies limited counting abilities of such encoders, recent work has shown that a slight extension of the hard attention mechanism can help in recognizing languages outside $AC^0$ (Hao et al., 2022). Instead of using unique hard attention, this model uses *average hard attention* (AHAT), which refers to the idea that the attention mechanism returns the uniform average value among all positions that maximize the attention. *To what extent does AHAT enrich the counting ability of UHAT?* In answering this question, we introduce a logic named $LTL(\mathbf{C}, +)$, which is an extension of LTL(Mon) that naturally incorporates counting features. We show that any language that can be defined within $LTL(\mathbf{C}, +)$ can also be identified by an AHAT. The logic $LTL(\mathbf{C}, +)$ can express interesting languages lying outside $AC^0$ including majority and parity. More generally, our result implies that AHATs are equipped with a powerful counting ability: all permutation-closed languages over a binary alphabet and all permutation closures of regular languages (which are in general not context-free) can be recognized by AHATs.

As a corollary, we provide a characterization of the "counting properties" of regular languages which can be captured by UHAT and AHAT. Two approaches for understanding counting properties of regular languages can be found in formal language theory: (1) Parikh-equivalence, and (2) taking letter-permutation closure. Both approaches "remove" the ordering from the input string, and as a result only the letter-count (more popularly called *Parikh images*) of an input string matters. In the setting of (1) (e.g. see (Parikh, 1966; Kozen, 1997)), a machine is allowed to *try all* letter reorderings of the input string $w$, and accepts iff the machine accepts *some* letter reordering of $w$. According to the well-known Parikh's Theorem (Parikh, 1966), each context-free language can in this way be captured by a regular language, e.g., $\{0^n 1^n : n \geq 0\}$ can be captured by $(01)^*$. We show in this paper that each regular language is Parikh-equivalent to an UHAT language, despite the fact that PARITY is not in UHAT. In the setting of (2), a machine must accept *all* letter permutations of an input string $w$. The letter-permutation closure of a regular language is not necessarily regular, e.g., such a closure language of $(abc)^*$ consists of all strings with the same number of $a$'s, $b$'s, and $c$'s, which is not even context-free. In this setting, although UHAT cannot capture regular languages like PARITY, AHAT can surprisingly capture all regular languages.

**Related work.** There has been very little research on identifying logical languages that can be accepted by transformers. The only example we are aware of is the recent work by Chiang et al. (2023), in which a variant of first-order logic with counting quantifiers is demonstrated to be embeddable into transformer encoders with a *soft attention* mechanism. The primary distinction between their work and our results is the choice of the attention mechanism. Additionally, the logic examined in their paper does not have access to the underlying word order being considered. This implies that some simple languages, such as $a^*b^*$, which are definable in FO, are not definable in their logic.

Due to the space constraints, some of the proofs are omitted and can be found in the online version of this paper (Barceló et al., 2023).

## 2 BACKGROUND NOTIONS AND RESULTS

### 2.1 TRANSFORMER ENCODERS

An *encoder layer* is a function that takes a sequence of vectors, $\mathbf{v}_0, \ldots, \mathbf{v}_{n-1}$, in $\mathbb{R}^d$ as input, where $d \geq 0$. It produces an output sequence of vectors of the same length, $\mathbf{v}'_0, \ldots, \mathbf{v}'_{n-1}$, in $\mathbb{R}^e$, with $e \geq 0$. The length of the sequence, $n$, can be arbitrary, but input and output dimensions, $d$ and $e$, are fixed for an encoder layer. For the first part of the paper we employ a *unique hard* attention mechanism, meaning that a position only attends to the element with the highest attention score.

Formally, an **encoder layer with unique hard attention** is given by two affinte transformations $A, B \colon \mathbb{R}^d \to \mathbb{R}^d$ and one feed-forward neural network $\mathcal{N} \colon \mathbb{R}^{2d} \to \mathbb{R}^e$ with ReLU activation function. For $i \in \{0, \ldots, n-1\}$, we set

$$\mathbf{a}_i \leftarrow \mathbf{v}_{j_i},$$

where $j_i \in \{0, \ldots, n-1\}$ is the minimum element that maximizes the *attention score* $\langle A\mathbf{v}_i, B\mathbf{v}_j \rangle$ over $j \in \{0, \ldots, n-1\}$. The $a_i$s are often known as *attention vectors*. After that, we set

$$\mathbf{v}'_i \leftarrow \mathcal{N}(\mathbf{v}_i, \mathbf{a}_i), \qquad i = 0, \ldots, n-1.$$

It is well-known that feed-forward neural networks with ReLU activation can express the function $\max\{x, y\}$. Thus, we may assume that $\mathcal{N}$ can be an arbitrary composition of affine functions with max.

**Transformer encoder.** A *unique hard attention transformer encoder* (UHAT)[1] is defined simply as the repeated application of encoder layers with unique hard attention.

### 2.2 LANGUAGES ACCEPTED BY TRANSFORMER ENCODERS

Next, we define how a transformer can be used to accept languages over a finite alphabet. This requires extending transformer encoders with three features: a function for representing alphabet symbols as vectors (which, for the purposes of this paper, we represent as one-hot encodings), another function that provides information about the absolute positions of these symbols within the input word, and a vector that is used for checking whether the word should be accepted or not. The function that provides information about positions is often referred to as a *positional encoding*, and it is essential for recognizing properties of ordered sequences of vectors. In fact, without positional encoding, encoders treat input sequences as invariant to permutations (Pérez et al., 2021).

Consider a finite alphabet $\Sigma$ and let $T$ be an UHAT that takes a sequence of vectors over $\mathbb{R}^d$ as input and converts it into a sequence of vectors over $\mathbb{R}^e$. A language $L \subseteq \Sigma^+$ is *accepted* by $T$, if there is an embedding function $f \colon \Sigma \to \mathbb{R}^d$, a positional encoding function $p \colon \mathbb{N} \times \mathbb{N} \to \mathbb{R}^d$, and a vector $\mathbf{t} \in \mathbb{R}^e$, such that for every $\bar{w} \in L$ we have $T'(\bar{w}) > 0$, and for every $w \in \Sigma^+ \setminus L$ we have $T'(\bar{w}) < 0$. Here, $T' \colon \Sigma^+ \to \mathbb{R}$ is defined as follows. Let $\bar{w} = a_0 \ldots a_{n-1} \in \Sigma^n$, and suppose the output of $T$ when given the input sequence $f(a_0) + p(0, n), \ldots, f(a_{n-1}) + p(n-1, n)$ is the sequence $\mathbf{v}_0, \ldots, \mathbf{v}_{n-1}$. Then we set $T'(\bar{w}) = \langle \mathbf{t}, \mathbf{v}_0 \rangle$.

---

[1] Some of the previous papers, for instance Hao et al. (2022), allow to use in UHAT only rational numbers. We find this too restrictive because functions such as cos and sin are widely used in practice. Nevertheless, we stress that our results hold with this restriction, by taking good-enough approximations by rational numbers.

## 2.3 First order logic on words

We assume familiarity with first-order logic (FO). Let $\Sigma$ be a finite alphabet. A word $\bar{w} = a_0 \cdots a_{n-1}$ in $\Sigma^+$ is represented as a structure $S_{\bar{w}}$ whose domain is $\{0, \ldots, n-1\}$. This structure includes a binary relation $<$ that is interpreted as the linear order on the domain, and for each symbol $a \in \Sigma$, there is a unary relation $P_a$ containing positions $i = 0, \ldots, n-1$ where $a_i = a$. Given an FO *sentence* over words, that is, an FO formula without free variables, we denote the language of all words $\bar{w} \in \Sigma^+$ satisfying $S_{\bar{w}} \models \phi$ as $L(\phi)$. If an $L \subseteq \Sigma^+$ satisfies $L = L(\phi)$, for some FO sentence $\phi$, then we say that $L$ is *definable in* FO.

**Example 1.** First-order logic (FO) enables us to define certain languages of interest. Here, we present an illustrative example. Initially, we recognize that we can employ FO to define a relation $\mathsf{first}(x) := \neg \exists y (y < x)$ that exclusively holds true at the first position of a word. Correspondingly, we can define a relation $\mathsf{last}(x) := \neg \exists y (x < y)$ that holds solely at the last position of the word. Moreover, it is possible to define a binary relation $\mathsf{succ}(x, y) := x < y \wedge \neg \exists z (x < z \wedge z < y)$, which defines the successor relation within the domain. With these expressions, we can show that FO is capable of defining the language $(ab)^+$:

$$\exists x \big(\mathsf{first}(x) \wedge P_a(x)\big) \ \wedge \ \exists x \big(\mathsf{last}(x) \wedge P_b(x)\big) \ \wedge \ \forall x \forall y \big(\mathsf{succ}(x, y) \rightarrow (P_a(x) \leftrightarrow P_b(y))\big).$$

That is, the first symbol of the word is an $a$, the last one is a $b$, every $a$ is followed by a $b$, and every $b$ is preceded by an $a$. □

## 2.4 Unary numerical predicates

It is known that FO sentences can only define regular languages. In turn, there are regular languages that are not definable in FO. An example is the language $(aa)^*$, which contains those words formed solely by the symbol $a$ that are of even length. However, there is a straightforward extension of FO that can define this language: all we need to do is add unary predicate $\mathsf{even}(x)$, which holds true at position $i$ in a word if and only if $i$ is even. In fact, extending FO with the predicate $\mathsf{even}(x)$ allows us to define the language $(aa)^*$ using the following formula, which indicates that the last symbol in the word satisfies the unary predicate even: $\forall x P_a(x) \ \wedge \ \forall y(\mathsf{last}(y) \rightarrow \mathsf{even}(y))$.

The extension of FO with unary numerical predicates can then be useful for defining languages. We define a *unary numerical predicate* $\Theta$ as an infinite family of functions

$$\theta_n : \{0, \ldots, n\} \rightarrow \{0, 1\}, \qquad n > 0.$$

Given a word $\bar{w}$ in $\Sigma^+$ of length $n$, for $n > 0$, we have that the predicate $\Theta(x)$ holds in position $i$ in $\bar{w}$ if and only if $\theta_n(i) = 1$ (so far, we do not use the value of $\theta_n$ at $n$ as positions are numbered from 0 to $n-1$. We will use this value in Section 4). Notice that under our definition, the truth of a unary numerical predicate at position $i$ in the word $\bar{w}$ depends not only on $i$ but also on the length of the word $\bar{w}$. As we will explore further, this characteristic is advantageous for defining interesting languages in FO extended with arbitrary unary numerical predicates. Following the literature, we write FO(Mon) for such an extension (Barrington et al., 2005).

**Example 2.** Consider, for example, the non-regular language $\{a^n b^n \mid n > 0\}$. We show that it can be expressed in FO(Mon) with the help of a unary numerical predicate $\Theta(x)$ such that $\theta_n(i) = 1$ iff $n$ is even and $i = n/2 - 1$. In fact, it suffices to use the formula: $\exists x \big(\Theta(x) \ \wedge \ P_a(x) \ \wedge \ \forall y(y < x \rightarrow P_a(y)) \ \wedge \ \forall y(x < y \rightarrow P_b(y))\big)$. This formula expresses that the middle point $i$ of $\bar{w}$ exists, is labeled as $a$, and all positions smaller than $i$ are also labeled $a$, while all positions larger than $i$ are labeled as $b$. This example illustrates the significance of unary numerical predicates depending on both the position and the length of the word over which the formula is evaluated. □

The definition of the language $L(\phi) \subseteq \Sigma^+$ defined by an FO(Mon) sentence $\phi$ is analogous to the one we provided for FO.

# 3 $\mathsf{AC}^0$ languages accepted by UHATs

## 3.1 Not all languages in $\mathsf{AC}^0$ are accepted by UHATs.

Hao et al. (2022) proved that languages accepted by UHATs belong to the circuit complexity class $\mathsf{AC}^0$, i.e., the class of languages accepted by families of Boolean circuits of unbounded fan-in,

constant depth, and polynomial size. We combine results by Ajtai (1983) and Hahn (2020) to show that the opposite is not the case, i.e., there are $AC^0$ languages that are not accepted by UHATs.

As shown in Ajtai (1983), there is an $AC^0$-family of circuits $\{C_n : \{0,1\}^n \to \{0,1\}\}_{n \in \mathbb{N}}$ such that for all $n$, the circuit $C_n$ accepts all strings with at at least $2n/3$ ones and rejects all strings with at most $n/3$. Consider a language *approximate majority*, consisting of strings accepted by circuits from $\{C_n\}$. This language is in $AC^0$ by construction. However, as we state next, it cannot be recognized by an UHAT. This result is proved by using a property of UHATs established in Hahn (2020).

**Proposition 1.** *There is no UHAT that accepts the language* approximate majority.

Viola (2009) shows that $\{C_n\}$ can be made polynomial-time computable, which implies the existence of a *polynomial-time computable* language from $AC^0$ that cannot be accepted by an UHAT.

### 3.2 MAIN RESULT: FO(Mon) LANGUAGES ARE ACCEPTED BY UHATS

Proposition 1 tells us that not all $AC^0$ languages are accepted by UHATs. In this section, we identify a significant subset of $AC^0$ languages that can be accepted by UHATs. To accomplish this, we rely on the characterization of the class $AC^0$ as those languages that can be defined in FO extended with arbitrary numerical predicates. Our main result establishes that as long as we restrict ourselves to unary numerical predicates, translation into UHATs is possible.

**Theorem 1.** *Let $\Sigma$ be a finite alphabet and $\phi$ an* FO(Mon) *sentence over words from the alphabet $\Sigma$. There is an UHAT that accepts $L(\phi)$.*

Proving this result by induction on FO(Mon) formulas, which would be the most natural approach to tackle the problem, turns out to be difficult. The challenge arises because the FO(Mon) formulas obtained by induction can have arbitrary arity, and transformer encoders do not seem capable of handling the requirements imposed by such formulas. To address this issue, we take a different approach. We employ Kamp's Theorem, which establishes that the languages definable in FO are precisely those that are definable in *linear temporal logic* (LTL) (Kamp, 1968).

### 3.3 USING LTL(Mon) TO PROVE OUR MAIN RESULT

We first explain how LTL is defined, as this is crucial to understanding the remainder of the paper. Let $\Sigma$ be a finite alphabet. LTL formulas over $\Sigma$ are defined as follows: if $a \in \Sigma$, then $a$ is an LTL formula. Additionally, LTL formulas are closed under Boolean combinations. Finally, if $\phi$ and $\psi$ are LTL formulas, then $\mathbf{X}\phi$ and $\phi\mathbf{U}\psi$ are also LTL formulas. Here, $\mathbf{X}$ is referred to as the *next* operator, and $\mathbf{U}$ as the *until* operator.

LTL formulas are unary, i.e., they are evaluated over positions within a word. Let $\bar{w} = a_0 \cdots a_{n-1}$ be a word in $\Sigma^+$, and let $i = 0, \ldots, n - 1$. We define the satisfaction of an LTL formula $\phi$ over $\bar{w}$ at position $i$, written as $(\bar{w}, i) \models \phi$, inductively as follows (omitting Boolean combinations):

- $(\bar{w}, i) \models a$ if and only if $a = a_i$, for $a \in \Sigma$.
- $(\bar{w}, i) \models \mathbf{X}\phi$ if and only if $i < n - 1$ and $(\bar{w}, i + 1) \models \phi$. In other words, $\phi$ holds in the next position after $i$ (if such a position exists).
- $(\bar{w}, i) \models \phi\mathbf{U}\psi$ if and only if there exists a position $j = i, \ldots, n - 1$ for which $(\bar{w}, j) \models \psi$ and such that $(\bar{w}, k) \models \phi$ for every $k$ with $i \leq k < j$. That is, $\phi$ holds starting from position $i$ until the first position where $\psi$ holds (and a position where $\psi$ holds must exist).

We can extend LTL with unary numerical predicates in the same way we did it for FO. Formally, we define LTL(Mon) as the extension of LTL with every formula of the form $\Theta$, for $\Theta$ a unary numerical predicate. We write $(\bar{w}, i) \models \Theta$ to denote that $\theta_n(i) = 1$, where $n$ is the length of $\bar{w}$. If $\phi$ is an LTL(Mon) formula over $\Sigma$, we write $L(\phi)$ for the set of words $\bar{w} \in \Sigma^+$ with $(\bar{w}, 0) \models \phi$.

Kamp's Theorem establishes that for every FO sentence $\phi$ there exists an LTL formula $\psi$ such that $L(\phi) = L(\psi)$, and vice-versa. It is straightforward to see that this property extends to the logics FO(Mon) and LTL(Mon).

**Proposition 2.** *(Kamp, 1968) For every* FO(Mon) *sentence $\phi$ there exists an* LTL(Mon) *formula $\psi$ such that $L(\phi) = L(\psi)$, and vice-versa.*

Our proof of Theorem 1 is then derived directly from Proposition 2 and the following result.

**Proposition 3.** *Let $\Sigma$ be a finite alphabet and $\phi$ an* LTL(Mon) *formula defined over words from the alphabet $\Sigma$. There is an UHAT $T$ that accepts $L(\phi)$.*

Before proving this result, we make the following important remark regarding the positional encoding $p$ used by $T$ to accept $L(\phi)$. On a pair $(i, n) \in \mathbb{N} \times \mathbb{N}$ with $i < n$, we have that $p(i, n)$ is composed of elements $i$, $1/(i+1)$, $(-1)^i$, $\cos\left(\pi(1-2^{-i})/10\right)$, $\sin\left(\pi(1-2^{-i})/10\right)$, and $\theta_n(i)$, for every unary numerical predicate $\Theta$ mentioned in $\phi$.

*Proof of Proposition 3.* Let $\phi$ be a formula of LTL(Mon). We say that a UHAT *realizes $\phi$ position-wise* if, given a word $\bar{w} = a_0 \ldots a_{n-1} \in \Sigma^+$, the UHAT outputs a sequence:

$$\mathbb{I}\{(\bar{w}, 0) \models \phi\}, \ \mathbb{I}\{(\bar{w}, 1) \models \phi\}, \ \ldots \ , \ \mathbb{I}\{(\bar{w}, n-1) \models \phi\};$$

that is, a binary word indicating for which positions $\phi$ is true on $\bar{w}$ and for which is false. We show by structural induction that every LTL(Mon) formula is realizable position-wise by some UHAT.

Let us consider first the base cases. If $\phi = a$, for some $a \in \Sigma$, our goal is to obtain a sequence:

$$\mathbb{I}\{a_0 = a\}, \ \mathbb{I}\{a_1 = a\}, \ \ldots \ , \ \mathbb{I}\{a_{n-1} = a\}.$$

This can easily be achieved by using a one-hot encoding as the embedding function. In turn, if $\phi = \Theta$, for $\Theta$ a unary numerical predicate, then $\phi$ can be realized position-wise using the corresponding positional encoding $p(i, n) = \theta_n(i)$.

We continue with Boolean combinations. They can be implemented position-wise by compositions of affine transformation and $\max$ as follows: $\neg x = 1 - x$ ' and $x \vee y = \frac{\max\{2x-1, 2y-1\}+1}{2}$.

For the cases when our formula is of the form $\mathbf{X}\phi$ or $\phi\mathbf{U}\psi$, we need the following lemma.

**Lemma 1.** *There is an UHAT that transforms each $x_0, \ldots, x_{n-1} \in \{0, 1\}$ as follows:*

$$x_0, \ldots, x_{n-2}, x_{n-1} \mapsto x_0, \ldots, x_{n-2}, 0.$$

Let us assume now that our formula is of the form $\mathbf{X}\phi$. It is enough to design a unique hard attention layer in which attention is always maximized at the next position. More precisely, we construct an UHAT that outputs a sequence of vectors $\mathbf{v}_1, \ldots, \mathbf{v}_n \in \mathbb{R}^3$, and a linear transformation $A \colon \mathbb{R}^3 \to \mathbb{R}^3$, such that $\arg\max_{j \in \mathbb{N}} \langle A\mathbf{v}_i, \mathbf{v}_j \rangle = \{i + 1\}$, for $i = 0, \ldots, n - 2$. This will allow us to "send" $\mathbb{I}\{(\bar{w}, i + 1) \models \phi\} = \mathbb{I}\{(\bar{w}, i) \models \mathbf{X}\phi\}$ to the $i$th position, for $i = 0, \ldots, n - 2$. It only remains then to apply Lemma 1 to obtain $0 = \mathbb{I}\{(\bar{w}, n - 1) \models \mathbf{X}\phi\}$ at the last position.

Using our positional encoding and an affine position-wise transformation, we can obtain:

$$\mathbf{v}_i = \left( \cos\left( \frac{\pi(1 - 2^{-i})}{10} \right), \ \sin\left( \frac{\pi(1 - 2^{-i})}{10} \right), \ (-1)^i \cdot 10 \right).$$

Let $A$ be a linear transformation that inverts the sign of the third coordinate. Observe that:

$$\langle A\mathbf{v}_i, \mathbf{v}_j \rangle = \cos\left( \frac{\pi(2^{-i} - 2^{-j})}{10} \right) + (-1)^{i+j+1} \cdot 10.$$

We claim that, for a fixed $i$, this quantity is maximized at $j = i + 1$. First, those $j$s that have the same parity as $i$ (in particular, $j = i$) cannot achieve the maximum because the second term is $-10$. For $j$s with a different parity, we have $\langle A\mathbf{v}_i, \mathbf{v}_j \rangle = \cos\left(\pi(2^{-i}-2^{-j})/10\right) + 10$. Since all angles are in $[-\pi/10, \pi/10]$, this quantity is maximized when $|2^{-i} - 2^{-j}|$ is minimized. For $j < i$, the last quantity is at least $2^{-i}$, and for $j > i$, the minimum of this quantity is $2^{-i-1}$, achieved at $j = i + 1$.

Let us finally assume that our formula is of the form $\phi\mathbf{U}\psi$. Observe that the value of $\phi\mathbf{U}\psi$ at position $i$ can be computed as follows: we go to the right, starting from the $i$th position, until we see a position $j_i$ where either $\phi$ is false, or $\psi$ is true, or this is the last position. Then $\phi\mathbf{U}\psi$ holds at $i$ if and only if $\psi$ holds at $j_i$. That is, $(\bar{w}, i) \models$ if and only if $(\bar{w}, j_i) \models \psi$, where $j_i \in \{i, \ldots, n-1\}$ is the minimal position with $\tau(j) = 1$, where, in turn, the sequence $\tau$ is defined by $\tau(i) = \mathbb{I}\{(\bar{w}, i) \models \neg\phi \vee \psi\} \vee \mathbb{I}\{i = n - 1\}$, for $i = 0, \ldots, n - 1$. To compute the sequence $\tau$, we first compute $\phi \wedge \neg\psi$

position-wise (we can do that because we already have $\phi$ and $\psi$ at our disposal), then we add the conjunction with $\mathbb{I}\{i \neq n-1\}$ by Lemma 1, and then we take the negation of the resulting sequence.

To show the lemma, it is enough to create a unique hard attention layer, where for every position $i$ the attention is maximized at $j_i$. Using our positional encoding and the induction hypothesis, we can obtain a sequence of vectors $\mathbf{v}_1, \ldots, \mathbf{v}_n \in \mathbb{R}^4$ such that:

$$\mathbf{v}_i = \left( \cos\left( \frac{\pi(1 - 2^{-i})}{10} \right), \; \sin\left( \frac{\pi(1 - 2^{-i})}{10} \right), \; 1, \; \tau(i) \right).$$

Consider a linear transformation $B \colon \mathbb{R}^4 \to \mathbb{R}^4$ such that

$$B\mathbf{v}_i = \left( \cos\left( \frac{\pi(1 - 2^{-i})}{10} \right), \; \sin\left( \frac{\pi(1 - 2^{-i})}{10} \right), \; 10\tau(i), 0 \right).$$

Observe that $\langle \mathbf{v}_i, B\mathbf{v}_j \rangle = \cos\left( \frac{\pi(2^{-i} - 2^{-j})}{10} \right) + 10\tau(j)$. We claim that this expression is maximized at $j = j_i$. First, because of the last term in it, it cannot be maximized at $j$ with $\tau(j) = 0$. It remains to show that among the $j$s with $(\bar{w}, j) \not\models \phi$, this quantity is minimized on the minimal $j$ which is at least $i$. In fact, in this case we have $\langle \mathbf{v}_i, B\mathbf{v}_j \rangle = \cos\left( \frac{\pi(2^{-i} - 2^{-j})}{10} \right) + 10$. All the angles in question are in $[-\pi/10, \pi/10]$, so the cosine is maximized when $|2^{-i} - 2^{-j}|$ is minimized. Now, this absolute value is at least $2^{-i}$ when $j < i$. In turn, this absolute value is smaller than $2^{-i}$ for $j \geq i$, and it is the smaller the smaller is $j$, as required. $\square$

## 3.4 APPLICATIONS OF OUR MAIN RESULT

We show two applications of our main result. First, UHATs accept all regular languages in $\mathsf{AC}^0$. Second, UHATs are strictly more expressive than regular and context-free languages in terms of the acceptance of languages up to letter-permutation.

**Regular languages in $\mathsf{AC}^0$.** There is an important fragment of $\mathrm{FO}(\mathsf{Mon})$ which is interesting in its own right. This is the logic $\mathrm{FO}(\mathsf{Mod})$, i.e., the extension of FO with unary numerical predicates of the form $\mathsf{Mod}_p^r$, for $p > 1$ and $0 \leq r \leq p - 1$. We have that $\mathsf{Mod}_p^r(i) = 1$ if and only if $i \equiv r \,(\mathrm{mod}\, p)$. In fact, by using a characterization given in Barrington et al. (1992), one can show that the languages definable in $\mathrm{FO}(\mathsf{Mod})$ are precisely the regular languages within $\mathsf{AC}^0$. Then:

**Corollary 1.** *Let $L \subseteq \Sigma^+$ be a regular language in $\mathsf{AC}^0$. There is an UHAT that accepts $L$.*

**Recognizing regular languages up to letter-permutation.** Although not all regular languages are accepted by UHATs (e.g. *parity*), we can use Theorem 1 to show that, up to letter-permutation, UHAT is in fact strictly more powerful than regular and context-free languages.

To formalize our result, we recall the notion of semilinear sets and the Parikh image of a language. A *linear set* $S$ is a subset of $\mathbb{N}^d$ (for some positive integer $d$, called *dimension*) of the form $\mathbf{v}_0 + \sum_{i=1}^r \mathbf{v}_i \mathbb{N} := \{\mathbf{v}_0 + \sum_{i=1}^r k_i \mathbf{v}_i : k_1, \ldots, k_r \in \mathbb{N}\}$ for some vectors $\mathbf{v}_0, \ldots, \mathbf{v}_r \in \mathbb{N}^d$. A *semilinear set* $S$ over $\mathbb{N}^d$ is a finite union of linear sets over $\mathbb{N}^d$. Semilinear sets have a very tight connection to formal languages through the notion of the *Parikh image* a language $L$ (Parikh, 1966), which intuitively corresponds to the set of "letter-counts" of $L$. More precisely, consider the alphabet $\Sigma = \{a_1, \ldots, a_d\}$ and a language $L$ over $\Sigma$. For a word $w \in \Sigma$, let $|w|_{a_i}$ denotes the number of occurrences of $a_i$ in $w$. The *Parikh image* $\mathcal{P}(L)$ of $L$ is defined to be the set of tuples $\mathbf{v} = (|w|_{a_1}, \ldots, |w|_{a_d}) \in \mathbb{N}^d$ for some word $w \in L$. For example, if $L = \{a^n b^n : n \geq 0\}$ and $L' = (ab)^*$, then $\mathcal{P}(L) = \mathcal{P}(L')$. In this case, we say that $L$ and $L'$ are *Parikh-equivalent*. Note that $L'$ is regular, while $L$ is context-free but not regular. This is not a coincidence based on the celebrated Parikh's Theorem (cf. Parikh (1966), also see Kozen (1997)).

**Proposition 4** (Parikh (1966)). *The Parikh images of both regular and context-free languages coincide with semilinear sets.*

In other words, although context-free languages are strict superset of regular languages, they are in fact equally powerful up to letter-permutation. What about UHATs? We have that they are strictly more powerful than regular and context-free languages up to letter-permutation.

**Proposition 5.** *Each regular language has a Parikh-equivalent language accepted by an UHAT. In turn, there is an UHAT language with no Parikh-equivalent regular language.*

# 4 LANGUAGES BEYOND AC$^0$

Transformer encoders with unique hard attention can only recognize languages in AC$^0$, but a slight extension of the attention mechanism allows to recognize languages lying outside such a class (Hao et al., 2022). In this section, we show that in fact such an extended model can recognize all languages definable in a powerful logic that extends LTL with counting features. This logic can express interesting languages outside AC$^0$, such as *majority* and *parity*.

## 4.1 AVERAGE HARD ATTENTION

For the results in this section, we consider an extended version of transformer encoders that utilize an *average hard attention mechanism* (Pérez et al., 2021; Hao et al., 2022). Following the literature, we call these AHAT.

**Encoder layer with average hard attention.** As before, these layers are defined by two affine transformations, $A, B \colon \mathbb{R}^d \to \mathbb{R}^d$ and one feed-forward neural network $\mathcal{N} \colon \mathbb{R}^{2d} \to \mathbb{R}^e$ with ReLU activation function. For every $i \in \{0, \ldots, n-1\}$, we define $S_i$ as the set of positions $j \in \{0, \ldots, n-1\}$ that maximize $\langle A\mathbf{v}_i, B\mathbf{v}_j \rangle$. We then set

$$\mathbf{a}_i \leftarrow \Big( \sum_{j \in S_i} \mathbf{v}_j \Big)/|S_i|.$$

After that, we set $\mathbf{v}'_i \leftarrow \mathcal{N}(\mathbf{v}_i, \mathbf{a}_i)$, for each $i = 0, \ldots, n-1$. That is, attention scores under average hard attention return the uniform average value among all positions that maximize attention.

We also use *future positional masking* that allows us to take into account only positions up to $i$. If the future positional masking is used, the sets $S_i$ are defined as sets of positions $j \in \{0, 1, \ldots, i\}$ that maximize $\langle A\mathbf{v}_i, B\mathbf{v}_j \rangle$. Positional masks have been employed on several occasions in theoretical papers (Yao et al., 2021; Bhattamishra et al., 2020; Hao et al., 2022) as well as in practice, for example, for training GPT-2 (Radford et al., 2019).

## 4.2 LTL EXTENDED WITH COUNTING TERMS

We present here LTL(**C**, +), an extension of LTL(Mon) that allows us to define counting properties over words in a simple manner. This requires the introduction of *counting terms* as defined next.

**Counting terms.** Suppose $\phi$ is a unary formula. Then $\overleftarrow{\#\phi}$ and $\overrightarrow{\#\phi}$ are counting terms. The interpretation of these terms in position $i$ of a word $\bar{w}$ of length $n$ is defined as follows:

$$\overleftarrow{\#\phi}(\bar{w}, i) = |\{j \in \{0, \ldots, i\} \mid (\bar{w}, j) \models \phi\}|, \quad \overrightarrow{\#\phi}(\bar{w}, i) = |\{j \in \{i, \ldots, n-1\} \mid (\bar{w}, j) \models \phi\}|.$$

That is, $\overleftarrow{\#\phi}(\bar{w}, i)$ is the number of positions to the left of $i$ (including $i$) that satisfy $\phi$, while $\overrightarrow{\#\phi}(\bar{w}, i)$ is the number of positions to the right of $i$ (including $i$) that satisfy $\phi$. Notice that, for words of length $n$, counting terms take values in $\{0, 1, \ldots, n\}$.

**Counting formulas.** With counting terms and unary numerical predicates we can create new formulas in the following way. Let $\phi$ be a unary formula and $\Theta$ a unary numerical predicate. We define new formulas $\Theta(\overleftarrow{\#\phi})$ and $\Theta(\overrightarrow{\#\phi})$. The interpretation of such formulas on position $i$ of a word $\bar{w}$ of length $n$ is as follows:

$$(\bar{w}, i) \models \Theta(\overleftarrow{\#\phi}) \iff \theta_n(\overleftarrow{\#\phi}(\bar{w}, i)) = 1 \qquad (\bar{w}, i) \models \Theta(\overrightarrow{\#\phi}) \iff \theta_n(\overrightarrow{\#\phi}(\bar{w}, i)) = 1.$$

That is, the number of positions to the left (resp., right) of $i$ (including $i$) that satisfy $\phi$ satisfies the predicate $\Theta$. As counting terms can take value $n$, the value of $\theta_n$ on $n$ becomes useful.

We also incorporate into our logic the possibility of checking linear inequalities with integer coefficients over counting terms. More specifically, for any finite set of unary formulas $\phi_1, \ldots, \phi_k, \psi_1, \ldots, \psi_k$, and for any coefficients $c_1, \ldots, c_k, d_1, \ldots, d_k \in \mathbb{Z}$ we can create a formula: $\sum_{j=1}^{k} c_j \cdot \overleftarrow{\#\phi_j} + \sum_{j=1}^{k} d_j \cdot \overrightarrow{\#\psi_j} \geq 0$, which is interpreted as follows:

$$(\bar{w}, i) \models \sum_{j=1}^{k} c_j \cdot \overleftarrow{\#\phi_j} + \sum_{j=1}^{k} d_j \cdot \overrightarrow{\#\psi_j} \geq 0 \iff \sum_{j=1}^{k} c_j \cdot \overleftarrow{\#\phi_j}(\bar{w}, i) + \sum_{j=1}^{k} d_j \cdot \overrightarrow{\#\psi_j}(\bar{w}, i) \geq 0.$$

**The logic** $\text{LTL}(\mathbf{C}, +)$**.** We denote by $\text{LTL}(\mathbf{C}, +)$ the logic that is recursively defined as follows:

- Every formula $\text{LTL}(\mathsf{Mon})$ is also an $\text{LTL}(\mathbf{C}, +)$ formula.
- Boolean combinations of $\text{LTL}(\mathbf{C}, +)$ formulas are $\text{LTL}(\mathbf{C}, +)$ formulas.
- If $\phi$ and $\psi$ are $\text{LTL}(\mathbf{C}, +)$ formulas, then so are $\mathbf{X}\phi$ and $\phi\mathbf{U}\psi$.
- If $\phi$ is an $\text{LTL}(\mathbf{C}, +)$ formula and $\Theta$ is a unary numerical predicate, then $\Theta(\overleftarrow{\#\phi})$ and $\Theta(\overrightarrow{\#\phi})$ are $\text{LTL}(\mathbf{C}, +)$ formulas.
- If $\phi_1, \dots, \phi_k, \psi_1, \dots, \psi_k$ are formulas of $\text{LTL}(\mathbf{C}, +)$, then $\sum_{j=1}^{k} c_j \cdot \overleftarrow{\#\phi_j} + \sum_{j=1}^{k} d_j \cdot \overrightarrow{\#\psi_j} \geq 0$, is a formula of $\text{LTL}(\mathbf{C}, +)$.

### 4.3 $\text{LTL}(\mathbf{C})$ definable languages are accepted by encoders

Next, we state the main result of this section: languages definable by $\text{LTL}(\mathbf{C}, +)$ formulas are accepted by transformer encoders with average hard attention.

**Theorem 2.** *Let $\Sigma$ be a finite alphabet and $\phi$ an $\text{LTL}(\mathbf{C}, +)$ formula defined over words from the alphabet $\Sigma$. There is an AHAT $T$ that accepts $L(\phi)$.*

As a corollary to Theorem 2, we show that AHATs are rather powerful in counting. To make this claim more formal, we study *permutation-closed* languages, i.e., languages $L$ such that $\bar{v} \in L$ iff any letter-permutation of $\bar{v}$ is in $L$. For a language $L$, we write $perm(L)$ to be the permutation-closure of $L$, i.e., $perm(L) = \{\bar{w} : \mathcal{P}(\bar{w}) = \mathcal{P}(\bar{v}), \text{ for some } \bar{v} \in L\}$. Observe that $perm((abc)^*)$ consists of all strings with the same number of occurrences of $a$, $b$, and $c$; this is not even context-free. Owing to Parikh's Theorem, to recognize $perm(L)$, where $L$ is a regular language, an ability to perform letter-counting and linear arithmetic reasoning (i.e. semilinear set reasoning) is necessary. AHATs possess such an ability, as shown by the following corollary.

**Corollary 2.** *The permutation closure $perm(L)$ of any regular language $L$ is accepted by an AHAT. Moreover, any permutation-closed language over a binary alphabet is accepted by an AHAT.*

Both *majority* and *parity* are permutation-closed and are over a binary alphabet. Hence, by the previous result, they are both accepted by AHATs. While for *majority* this was known (Hao et al., 2022), the result for *parity* is new.

## 5 Conclusions and future work

We have conducted an investigation of the problem of which languages can be accepted by transformer encoders with hard attention. For UHATs, we have demonstrated that while they cannot accept all languages in $\text{AC}^0$, they can still accept all languages in a 'monadic' version of it defined by the logic $\text{FO}(\mathsf{Mon})$. Crucial to the proof of this result is the equivalence between FO and LTL, as provided by Kamp's Theorem. In turn, we have shown that AHATs are capable of expressing any language definable in a powerful counting logic, $\text{LTL}(\mathbf{C}, +)$, that can express properties beyond $\text{AC}^0$. This implies, among other things, that the *parity* language can be accepted by an AHAT.

In a work of Anglin et al. (2023), contemporaneous to ours, it was shown that the logic $\text{FO}(\mathsf{Mon})$ exactly captures languages, accepted by UHATs with positional masking and *finite-valued positional encoding*. At the same time, with our technique, it can be shown that there are languages beyond $\text{FO}(\mathsf{Mon})$ that can be accepted by UHATs with infinite-valued positional encoding, for instance, the language of *palindromes*. The idea is that since we do not use positional masking, we can use arbitrary order on positions, in particular, one that puts the language of palindromes into $\text{FO}(\mathsf{Mon})$. Due to the space constraints, we omit a more detailed discussion of this.

### Acknowledgments

Barceló and Kozachinskiy are funded by ANID–Millennium Science Initiative Program - ICN17002 and by the National Center for Artificial Intelligence CENIA FB210017, Basal ANID. Anthony Lin was supported by European Research Council under European Union's Horizon research and innovation programme (grant agreement no 101089343).

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
