APPENDIX

*Proof of Proposition 1.* As Hahn showed, for every $\varepsilon > 0$ and $L > 0$ there exists $c \geq 0$ such that, for all larger enough $n$, if we consider as inputs binary strings of length $n$, for every UHAT $T$ consisting of $L$ layers, there exists a fixation of $\varepsilon n$ input bits such that, under this fixation, the output of $T$ is determined by $c$ unfixed bits  Hahn (2020). However, it cannot hold for an UHAT recognizing *approximate majority*, for example, when $\varepsilon = 1/10$. Regardless of how we fix $n/10 + c$ input bits, if we fix the remaining bits to 0s, the circuit $C_n$ rejects our string, and if we fix them to 1s, it accepts our string, even though the output of the UHAT remains unchanged. □

*Proof of Lemma 1.* At position $i = 0, \ldots, n-1$, this transformation can be written as follows:

$$x_i \mapsto x_i - \max\{0, x_i + i - (n+1)\}.$$

It can easily be done with ReLU layer, using a positional encoding $p(i, n) = i - n$. However, it can also be done with a positional encoding that does not depend on $n$, for example $p(i) = (i, 1/(i+1))$. We just have to "transmit" $n-1$ to every position in the UHAT. For that, it is enough to have a unique hard attention layer, where attention in every position is maximized at $j = n-1$ (which allows that to "send" $n$ to every position). For instance, consider $\mathbf{v}_i = 1/(i+1)$, $A(x) = -x$, and observe that:

$$\arg \max_{j=0,\ldots,n-1} \langle A\mathbf{v}_i, \mathbf{v}_j \rangle = \arg \max_{j=0,\ldots,n-1} -\frac{1}{(i+1)(j+1)} = \{n-1\}$$

for every $i = 0, \ldots, n-1$. This finishes the proof of the lemma. □

*Proof of Proposition 5.* **Upper bound:** We first show that every regular language over $\Sigma = \{a_1, \ldots, a_d\}$ has a Parikh-equivalent language in UHAT. By Parikh's Theorem, the Parikh image of this given regular language is represented by a semilinear set $S$ in dimension $d$. Our proof employs Theorem 1. Since FO(Mon) is closed under disjunction, it suffices to consider only linear sets $S$. Thus, take an arbitrary linear set $S = \mathbf{v}_0 + \sum_{i=1}^r \mathbf{v}_i \mathbb{N}$, where $\mathbf{v}_i$ $(i > 0)$ is a non-zero vector. We will give a language $L$ over the alphabet of $\Sigma = \{a_1, \ldots, a_d\}$ definable in FO(Mon) (thus UHAT-recognizable, by Theorem 1) such that $\mathcal{P}(L) = S$. We will use the linear set $S = (1, 1, 0) + (2, 0, 1)\mathbb{N}$ as a running example.

For $i = 0, \ldots, r$ and $j = 1, \ldots, d$, define $v_i^j$ to be the natural number corresponding to the $j$th argument of $\mathbf{v}_i$. Define $w_i^j$ to be the string $a_j^{v_i^j}$, i.e., $a_j$ repeated $v_i^j$ times, while $\ell_i$ denotes the "length abstraction" of $\mathbf{v}_i$, i.e., $\ell_i := \sum_{j=1}^d v_i^j$. Finally, let $w_i$ be the concatenation of $w_i^1, \ldots, w_i^d$. Using our example of $S = (1, 1, 0) + (2, 0, 1)\mathbb{N}$, then we have $w_0 = a_1 a_2$ and $w_1 = a_1 a_1 a_3$. We also have $\ell_0 = 2$ and $\ell_1 = 3$.

Next we define the language $L$ as follows:

$$L := w_0 \cdot w_1^* \cdots w_r^*$$

Using our running example, $L$ would be $a_1 a_2 (a_1 a_1 a_3)^*$. It is easy to see that $\mathcal{P}(L) = S$.

To show that this language is in FO(Mon)-definable, we demonstrate that it is regular and belongs to $\mathsf{AC}^0$. It is regular because it is defined through concatenation and Kleene star. Since $\mathsf{AC}^0$ is closed under concatenation[2] it remains to show that languages of the form $w^*$, where $w$ is a word, are in $\mathsf{AC}^0$. We only have to care about input lengths that are multiples of $|w|$, for other input lengths the language is empty. Then we split the input into blocks of $|w|$ letters. We just need an $\mathsf{AC}^0$-circuit, checking that every block coincides with $w$. For example, this can be done with an AND over blocks of constant-size circuits, checking equality to $w$.

**Lower bound:** An example of a language that is in FO(Mon) (and so in UHAT) whose Parikh image is not semilinear (and therefore, no Parikh-equivalent regular language) is

$$L = \{a^k : k \text{ is a prime number}\}.$$

Note that $\Sigma = \{a\}$. This can be easily defined in FO(Mon) using the unary predicate $\Theta := \{k \in \mathbb{N} : k+1 \text{ is a prime number}\}$ as follows: $\exists x \Theta(x) \wedge \neg \exists y > x$. □

---

[2]if we have $\mathsf{AC}^0$-circuits $C_1, C_2$ for languages $L_1, L_2$, we can construct an $\mathsf{AC}^0$-circuit $C$ for their concatenation as follows: $C(x_1 \ldots x_n) = \bigvee_{i=1,\ldots,n} (C(x_1 \ldots x_i) \wedge C(x_{i+1} \ldots x_n))$.

*Proof of Theorem 2.* As before, we are proving that every formula $\phi$ of LTL$(\mathbf{C}, +)$ can be computed position-wise by some AHAT encoder, via structural induction. We have already shown how to do induction for all operators of LTL(Mon). In our proof, attention was always maximized at the unique $j$, and in this case, there is no difference between unique and average hard attention.

It remains to show the same for operators that are in LTL$(\mathbf{C}, +)$ but not in LTL(Mon). First, we show that given a formula $\phi$, computed position-wise by some AHAT, there is also an AHAT that computes $\overleftarrow{\#\phi}$ and $\overrightarrow{\#\phi}$ position-wise.

Using future positional masking and equal weights, we can compute at position $i$ the quantity:

$$y_i = \frac{\phi(w, 0) + \ldots + \phi(w, i)}{i + 1} = \frac{\overleftarrow{\#\phi}(w, i)}{i + 1}, \qquad i = 0, 1, \ldots, n - 1.$$

Next, we have to compute

$$z_i = \frac{\left(\overleftarrow{\#\phi}(w, i) - \phi(w, i)\right)}{i + 1}.$$

This can be achieved as follows:

$$z_i = y_i - \frac{\phi(w, i)}{i + 1} = y_i - \min\left\{\phi(w, i), \frac{1}{i + 1}\right\}.$$

As our positional encoding includes $1/(i+1)$, this computation is a composition of ReLU and affine transformations.

Our next goal is to get rid of the coefficient $1/(i + 1)$. For that, we create a layer with the following attention function:

$$\langle A\mathbf{v}_i, B\mathbf{v}_j \rangle = 2j \cdot z_i - \frac{j^2}{i + 1}, \qquad i, j = 0, \ldots, n - 1. \tag{1}$$

Such attention function is possible because (1) is a bilinear form of $\mathbf{v}_i$ and $\mathbf{v}_j$. Indeed, $\mathbf{v}_i$ contains $1/(i + 1)$ and $\mathbf{v}_j$ contains $j, j^2$ due to our positional encoding, and also $\mathbf{v}_i$ contains $z_i = \frac{\left(\overleftarrow{\#\phi}(w, i) - \phi(w, i)\right)}{i + 1}$.

Denoting $d_i = \overleftarrow{\#\phi}(w, i) - \phi(w, i)$, we get that (1) is equal to

$$\langle A\mathbf{v}_i, B\mathbf{v}_j \rangle = 2j \cdot \frac{d_i}{i + 1} - \frac{j^2}{i + 1} = \frac{-(d_i - j)^2 + d_i^2}{i + 1}.$$

Observe that $d_i = \overleftarrow{\#\phi}(w, i) - \phi(w, i)$ takes values in $\{0, \ldots, n - 1\}$. Hence, for a fixed $i$, the quantity (1) is uniquely maximized at $j = d_i$. In this way, we get $j = d_i$ to position $i$. Adding $\phi(w, i)$ to $d_i$, we get $\overleftarrow{\#\phi}(w, i)$. To get $\overrightarrow{\#\phi}(w, i)$ to position $i$, we observe that:

$$\overrightarrow{\#\phi}(w, i) = (\phi(w, 0) + \ldots + \phi(w, n - 1)) - ((\phi(w, 0) + \ldots + \phi(w, i - 1)))$$
$$= \overleftarrow{\#\phi}(w, n - 1) - d_i.$$

This is computable at position $i$ because $\overleftarrow{\#\phi}(w, n - 1)$ can be "sent" to all positions via the attention function, always maximized at the last position (see the proof of Lemma 1).

Our next goal is: given a formula $\phi$, computable position-wise by some AHAT, and a unary numerical predicate $\Theta$, provide an AHAT that computes $\Theta(\overleftarrow{\#\phi})$ and $\Theta(\overrightarrow{\#\phi})$ position-wise. As we have already shown, we can assume that we already have counting terms $\overleftarrow{\#\phi}$ and $\overrightarrow{\#\phi}$ computed position-wise. Next, we create a layer with the following attention function:

$$\langle A\mathbf{v}_i, B\mathbf{v}_j \rangle = 2j \cdot \overrightarrow{\#\phi}(w, i) - j^2 = -(j - \overrightarrow{\#\phi}(w, i))^2 + \overrightarrow{\#\phi}(w, i)^2.$$

Again, this is possible because this expression is a bilinear form of $\mathbf{v}_i$ and $\mathbf{v}_j$, due to our positional encoding. It is maximized at $j_i = \min\{n - 1, \overrightarrow{\#\phi}(w, i)\}$ (when the counting term is equal to $n$,

since we do not have a position indexed by $n$, the maximizing position will be $j_i = n - 1$). Having $\Theta$ included in the positional encoding, we can get $j_i$ and $\theta_n(j_i)$ to the $i$th position. Observe that:

$$\theta_n(\overrightarrow{\#\phi}(w, i)) = (\mathbb{I}\{\overrightarrow{\#\phi}(w, i) \leq n - 1\} \wedge \theta_n(j_i)) \vee (\neg\mathbb{I}\{\overrightarrow{\#\phi}(w, i) \leq n - 1\} \wedge \theta_n(n))$$

Since in our positional encoding, $\theta_n(n)$ is included in every position, and since position-wise Boolean operations can be done by an AHAT, it remains to compute the indicator $\mathbb{I}\{\overrightarrow{\#\phi}(w, i) \leq n - 1\}$. Transmitting $n$ once again to every position, we can write:

$$\mathbb{I}\{\overrightarrow{\#\phi}(w, i) \leq n - 1\} = \min\{1, n - \overrightarrow{\#\phi}(w, i)\}.$$

This quantity can be computed by a composition of ReLU and affine transformations. We can get $\theta_n(\overleftarrow{\#\phi}(w, i))$ to the $i$th position analogously.

Finally, we have to check that linear inequalities over counting terms can be done in AHAT. Given formulas $\phi_1, \ldots, \phi_k, \psi_1, \ldots, \psi_k$ already computed position-wise by some AHAT, we have to provide an AHAT that computes the formula $\sum_{j=1}^{k} c_j \cdot \overleftarrow{\#\phi_j} + \sum_{j=1}^{k} d_j \cdot \overrightarrow{\#\psi_j} \geq 0$ position-wise. After computing counting terms for $\phi_1, \ldots, \phi_k, \psi_1, \ldots, \psi_k$, we first can compute their linear combination, using affine position-wise transformations:

$$l_i = \sum_{j=1}^{k} c_j \cdot \overleftarrow{\#\phi_j}(w, i) + \sum_{j=1}^{k} d_j \cdot \overrightarrow{\#\psi_j}(w, i).$$

Since coefficients are integral, $l_i$ is integral as well, so we get:

$$\mathbb{I}\{l_i \geq 0\} = \max\{\min\{0, l_i\} + 1, 0\}.$$

The last expression can be computed via composition of ReLU and affine transformations.

$\square$

*Proof of Corollary 2.* We show that permutation-closed languages over binary alphabets and languages of the form $perm(L)$, where $L$ is a regular language, are expressible in $\mathrm{LTL}(\mathbf{C}, +)$.

First, assume that $L$ is a permutation-closed language over a binary alphabet $\{a, b\}$. Then whether or not a word $\bar{w}$ belongs to $L$ is determined by the length of $w$ and the number of $a$'s in $w$. In other words, there a numerical predicate $\Theta$ such that for every $n$ and for every $\bar{w} \in \{a, b\}^n$, we have $\bar{w} \in L$ if and only if $\theta_n(|\bar{w}|_a) = 1$ (recall that for a word $\bar{w}$ and for a letter $a$, the expression $|\bar{w}|_a$ denotes the number of occurrences of $a$ in $\bar{w}$). Thus, $L$ is expressible by the formula $\Theta(\overrightarrow{\#a})$.

We now show that every language of the form $perm(L)$, where $L$ is regular, is expressible in $\mathrm{LTL}(\mathbf{C}, +)$.

As shown in Parikh (1966), if $L$ is a regular language over the alphabet $\Sigma = \{a_1, \ldots, a_d\}$, then

$$perm(L) = \{w : \mathcal{P}(w) \in S\},$$

for some semilinear set $S$ of dimension $d$. Semilinear sets correspond precisely to sets of tuples that are definable in Presburger Arithmetic (e.g. see Haase (2018)). See standard textbook in mathematical logic for more details on Presburger Arithmetic (e.g. see Anderton (2001)). Since Presburger Arithmetic admits quantifier-elimination, we may assume that $S$ is a boolean combination of (a) inequalities of linear combination of counting terms, and (b) modulo arithmetic on counting terms (i.e. an expression of the form $|w|_{a_i} \equiv k \pmod{c}$, for some concrete natural numbers $0 \leq k < c$ and $c > 0$). For (b), one simply handles this using the formula $\Theta(\overrightarrow{\#a_i})$, where $\Theta$ is a unary numerical predicate consisting of all numbers $n$ such that $n \equiv k \pmod{c}$. For (a), take a linear inequality of the form

$$\psi(|w|_{a_1}, \ldots, |w|_{a_d}) := \sum_{i=1}^{d} c_i |w|_{a_i} \geq 0,$$

where $c_1, \ldots, c_d \in \mathbb{Z}$. Such a formula $\psi$ is already an atom permitted in $\mathrm{LTL}(\mathbf{C}, +)$. Since $\mathrm{LTL}(\mathbf{C}, +)$ is closed under boolean combination, it follows that $perm(L)$ is also in $\mathrm{LTL}(\mathbf{C}, +)$ and therefore, by Theorem 2, is in AHAT. $\square$