# OpenReview forum: "Logical Languages Accepted by Transformer Encoders with Hard Attention"
_ICLR.cc/2024/Conference — ICLR 2024 poster_

### Official Review · Reviewer_yxZd · 2023-10-30

**Soundness:** 4 excellent
**Presentation:** 4 excellent
**Contribution:** 3 good
**Rating:** 8
**Confidence:** 3

**Summary:**

This paper theoretically analyses the formal languages that transformer encoders can recognize.
The paper analyses two classes of transformers depending on the attention mechanism. The first class is
unique hard attention transformers (UHAT), and the second class is average hard attention transformers (AHAT).
It is known that UHAT encoders can only recognize languages inside the circuit complexity class $AC^0$. AHAT
can recognize languages outside $AC^0$, but its expressive power still lies in the circuit complexity class $TC^0$.

This paper gives new theoretical results on the topic. The main findings of the paper are:
- There is a language in $AC^0$ that UHAT cannot recognize.
- UHAT can recognize all languages definable in first-order logic with arbitrary unary numerical predicates. The class includes all regular languages in $AC^0$.
- AHAT can recognize all languages definable by LTL(C, +), which is an extension of the logical language acceptable by UHAT with counting terms.

**Strengths:**

**Important topic, impressive results:**
 Since the transformer is one of the most important neural architectures, understanding its expressiveness in a formal way seems an important topic. This paper brings progress on this topic by relating the expressive power of transformers with logical languages.
This connection results in identifying some important language classes that can be acceptable by UHAT, e.g.,  all regular languages in $AC^0$.

**A clearly written paper:** The paper is very clearly written. I feel no difficulty in reading the paper.
The background needed to understand the contribution is concisely explained in the paper.
Proofs of important theorems are shown in the main body of the paper, and they are easy to follow.

**New approaches for proofs.**
The paper uses the relationship between first-order logic and linear-time logic to prove the main results. This technique seems not
used in the previous papers analyzing the expressive powers of transformer encoders.

**Weaknesses:**

Currently, I have no clear reason to reject the paper.

**Questions:**

On p.6, the paper says, "Observe that $(\bar{w}, i) \models \phi U \psi $ if and only if $(\bar{w}, j_i) \models \psi$." I wonder what happens if there exists $i \leq j^\prime < j_i$ such that $(\bar{w}, j^\prime) \models \psi$"? Following the definition on page 5, I think $(\bar{w}, i) \models \phi U \psi$ holds if there exists such $j^\prime$.


**Minor comments:**
- **Abstract:** outside AC^0) -> AC^0 ?
- **page2, before related work:** it have been shown before that parity can be accepted by an AHAT -> it has not been shown before that parity can be accepted by an AHAT?
- **p.5, first paragraph:** at at least 2n/3 -> at least 2n/3
- **Appendix, proof of lemma1:** max{0, x_i + i - (n+1)} -> max{0, x_i + 1 - (n-1)}?

---

> ### Author Response · Authors · 2023-11-19
>
> Many thanks for your nice comments about the paper!
>
> Regarding your question: You are right, there is a small problem with the way in which j_i is defined. The proof, however, works verbatim after a slight modification of the definition of j_i. Please see the modified pdf. Thanks for noticing this!

---

### Official Review · Reviewer_auhY · 2023-10-30

**Soundness:** 2 fair
**Presentation:** 2 fair
**Contribution:** 4 excellent
**Rating:** 8
**Confidence:** 3

**Summary:**

The papers is a contribution to the exciting and challenging domain of characterizing expressivity of transformers. The authors focus on the  Unique Hard Attention Transformers (UHAT). Previous results have shown that $UHAT \subseteq AC^{0}$, where $AC^{0}$ is the circuit complexity class of circuits with constant depth, with unlimited fan-in  $AND$, and $OR$ gates. The author's first result is to show that  $UHAT \subset AC^{0}$. The key contribution of the paper is that First Order Logic on words with unary numerical predicates (i.e. unary boolean functions on positions of the symbols in a word, and the length of the word) is readable by UHAT. They denote this fragment with FO(Mon), the proof strategy used by authors relies on a classic result in logic known as the Kamp's theorem, which says that FO(Mon) is equivalent to another type of logic, known as the Linear Temporal Logic (Mon) i.e. the extension of LTL with unary numerical predicates. The authors then design vector encodings and positional encodings that allow evaluation of any LTL (Mon) formula using a UHAT. Finally, they show the applications of their results by comparing them to other types of formal languages.

**Strengths:**

Formal properties of transformers are not very well-understood, and analyzing them w.r.t. the formal languages they accept is a very exciting and challenging direction. The author's provide significant contributions in this direction, and the paper contributes many new results and ideas that can contribute to theoretical investigation of transformers. Writing is quite clear, the proof though hard, seems to consist of clear arguments (I mention my confusions in the questions).

**Weaknesses:**

The proof on Page 6 and Page 7 could be further clarified. Although, the structural induction arguments are clear, but I am not sure how this is consistent with the meaning of accepting a word as part of the language --- which authors define earlier (See questions)

**Questions:**

- In my understanding, in section 2.2 paragraph 2, it is unclear to me why you set $T(\bar{w})$ to $\langle \mathbf{t},\mathbf{v}_{0}\rangle$.
- [Minor Comment] Page 6, "reverses the third coordinate" is not a very precise statement.
- When is the notion of $T(\bar{w}) > 0$ (as introduced in section 2.2), used as the criterion in proof on page 6 and page 7. From this proof, I just see that you can perform LTL operations on input strings, but I am not sure how this shows that a string in the language will never be mapped to a string outside the language?
- Does Kamp's theorem give any bounds on the length of equivalent FO(Mon) equivalent formula in LTL(Mon)?

---

> ### Author Response · Authors · 2023-11-19
>
> Many thanks for your comments! We provide answers to your questions below.
>
> Q: In my understanding, in section 2.2 paragraph 2, it is unclear to me why you set T(w) to T(w,v0).
>
> A: We define the acceptance of a language using the content of the first position of the last layer, more specifically, we take a linear function (with values in R) of the value in this position, and we require that this function is positive for words in the language, and negative for words not in the language.
>
> Q: Page 6, "reverses the third coordinate" is not a very precise statement.
>
> A: We meant ``multiplies the third coordinate by (-1)’’. We have restated this (see new version of the pdf).
>
> Q: When is the notion of T(w) >0 (as introduced in section 2.2), used as the criterion in proof on page 6 and page 7. From this proof, I just see that you can perform LTL operations on input strings, but I am not sure how this shows that a string in the language will never be mapped to a string outside the language?
>
> A: We show that for any LTL language, there is a transformer encoder that maps strings from the language to 1 and strings not from the language to 0 (when we look at the first position at the last layer). To satisfy our acceptance criterion, it is enough to make an additional affine position-wise transformation x \mapsto x - ½, which maps 1 to ½  and 0 to -½. We should have added this remark and we, of course, will do that.
>
> Q: Does Kamp's theorem give any bounds on the length of equivalent FO(Mon) equivalent formula in LTL(Mon)?
>
> A: Translations from FO to LTL are non-elementary (which is unavoidable given known results on the satisfiability problem for such logics over finite words). In general, if the quantifier depth of the FO formula A is k, then an equivalent LTL formula is of size k-th exponential on the size of A.

---

> ### Comment · Reviewer_auhY · 2023-11-22
> **Read the rebuttal**
>
> I have read the rebuttal from the authors and comments from other reviewers. I am inclined to keep my score, as I believe the investigated problem is interesting and authors have clarified all the concerns raised by other reviewers. However, through the discussions, clarity of the paper has emerged to be a major concern. I hope that authors will rectify all these concerns for the camera-ready version.

---

### Official Review · Reviewer_maoo · 2023-10-31

**Soundness:** 2 fair
**Presentation:** 3 good
**Contribution:** 2 fair
**Rating:** 3
**Confidence:** 3

**Summary:**

This paper studies the expressiveness of Transformer encoders by establishing their relation with a class of formal languages called circuits. Based on prior work, the main theoretical claims of the paper are:

Previous works have demonstrated that UHAT transformers (similar to Transformers with hard attention) cannot recognize languages beyond AC^0 (informally: circuits of polynomial size and constant depth). This paper further finds an example problem class within AC^0 that cannot be recognized by UHAT, which demonstrates that AC^0 is not a “lower bound” of UHAT.

The paper further establishes a class of problems that can be recognized by UHAT.

Results are then extended to AHAT (Transformers with averaged head attention).

**Strengths:**

Justifying the expressiveness of Transformers through the length of formal language is a very important topic and could lead to better understanding of the working mechanisms of Transformers. This paper strengthens prior theoretical results and better bounds the expressivness of UHAT and AHAT.

**Weaknesses:**

Although the theoretical results themselves sounds interesting, I found some definitions and assumptions are not approprately stated, which potentially leads to incorrect results. While it is possible that the theoretical results still hold after fixing all the problems, I think the paper needs a major revision to ensure its validity.

Missing important restrictions when defining the transformer model.

- Precision of the number processed by the transformer. The paper does not include any restriction on the precision of the numbers processed by the transformer. This could make the model unrealistically expressive as discussed in many related work (e.g., proving Turing completeness of RNNs require relaxations on the numerical precision). In related works, a realistic assumption could be log-precision transformers, i.e., the number of floating/fixed-point bits scale logarithmically with the sequence length.

- No assumptions have been made about the number of transformer layers. Prior work usually assume constant depth or logarithm depth (w.r.t. sequence length). Related to this assumption, it seems that the proof of Proposition 2 constructs a Transformer whose number of layers depends on the form of input LTL. This makes it particularly important to make the correct assumption.

- Structure of the model. The model does not include residual connections and only uses single-head attention. Also the ReLU layer only applies ReLU to a single element of the input vector. Although these might be able to adapt in the proof, it would still be nice to make these assumptions as practical as possible.

Many related works are missing. The paper states that “there has been very little research on identifying logical languages that can be accepted by transformers”. However, with a quick google scholar search, I found the following highly-related papers not cited in the paper. It would be nice to discuss the relation of this paper’s results with these prior works.

[1] Merrill, William, Ashish Sabharwal, and Noah A. Smith. "Saturated transformers are constant-depth threshold circuits." Transactions of the Association for Computational Linguistics 10 (2022): 843-856.

[2] Merrill, William, and Ashish Sabharwal. "The parallelism tradeoff: Limitations of log-precision transformers." Transactions of the Association for Computational Linguistics 11 (2023): 531-545.

[3] Strobl, Lena. "Average-Hard Attention Transformers are Constant-Depth Uniform Threshold Circuits." arXiv preprint arXiv:2308.03212 (2023).

**Questions:**

Some important assumptions seem to be missing.

It would be nice to discuss the relation between this paper and the missed related work mentioned in the weaknesses section.

---

> ### Author Response · Authors · 2023-11-19
>
> Thanks for your comments and questions. We provide responses to them below.
>
> Q: Precision of the number processed by the transformer. The paper does not include any restriction on the precision of the numbers processed by the transformer ...
>
> A:  Firstly, we would like to point out that even infinite precision cannot take UHAT outside AC^0, as shown by Hao et al. (2022).
>
> Our current positional encoding needs polynomially many bits of precision. In fact, we have an alternative construction where instead of cosines in the positional encoding we simply use 2^{-i} and 2^{-2i} in the ith position. For our construction we need an attention that, at position i, is strictly smaller for any j < i than for any j >= i, and moreover, which strictly decrease for j = i, i+1, … An example of such attention is -(2^{-i} - 2^{j})^2 which is a bilinear of form in the positional encoding mentioned above. It seems open how to achieve the same result with logarithmically many bits of precision.
>
> Q: No assumptions have been made about the number of transformer layers. Prior work usually assume constant depth or logarithm depth (w.r.t. sequence length). Related to this assumption, it seems that the proof of Proposition 2 constructs a Transformer whose number of layers depends on the form of input LTL. This makes it particularly important to make the correct assumption.
>
> A: Our Transformers are of constant depth wrt the sequence length. You are right that the depth of the transformer that represents a given LTL formula phi depends on the structure of phi. We will make this more explicit in the final version of the paper.
>
> Q: Structure of the model. The model does not include residual connections and only uses single-head attention. Also the ReLU layer only applies ReLU to a single element of the input vector ...
>
> A:  We could include these features into the model, but note that this would have made our main result – that transformers accept every language in FO(Mon) – only weaker.  This is because we are showing that even without these features transformers can still do that. As for the lower bound (that transformer encoders cannot accept every language in AC^0), it relies on the sensitivity technique of Hahn, which is valid in the case with constant number of attention heads and arbitrary activation and attention functions. In other words, the lower bound is quite robust and does not depend on the technical details of the model.
>
> As for the ReLU – it is true that in practice as activation one uses feed-forward networks, not just application of ReLU to a single element. Still, one might note that in the context of acceptance of formal language it does not matter – both variants just mean that position-wise, we can use arbitrary compositions of affine transformations and max-function.
>
> We agree, however, that it is better to elaborate more on how these assumptions actually look at practice, and we will definitely do that.

---

> > ### Comment · Reviewer_maoo · 2023-11-21
> > **Major concern regarding the depth assumption**
> >
> > I thank the authors for their detailed response. I need to re-evaluate the added assumptions in the paper, and I already have a further question regarding the depth assumption and the proof of Proposition 2. In the response, the authors say that they assume **constant** depth of the transformer. However, they then mentioned that "the depth of the transformer that represents a given LTL formula phi depends on the structure of phi". In this case, it seems that the depth cannot be upper-bounded by a **universal** constant.
> >
> > Maybe the authors meant that the depth is a constant **depending on the sequence length**. However, this is a much stronger assumption compared to related papers. They mostly assume either a universal constant depth or that the depth depends logarithmically w.r.t. the sequence length.
> >
> > Please let me know if my understanding is incomplete. Thanks.

---

> > > ### Author Response · Authors · 2023-11-21
> > >
> > > Thanks! Let us clarify what we meant by constant depth: Our proof shows that for every language L in FO(Mon) there is a constant c(L) and a UHAT A with c(L) layers such that, for every word w, it is the case that w belongs to L if, and only if, w is accepted by A (i.e. A(w) > 0).
> > >
> > > So, to clarify further: The constant c(L) *does not depend on the input length*, only on the language L (notice that this is absolutely standard when obtaining results of this sort: see, e.g., https://arxiv.org/pdf/2310.13897.pdf and https://arxiv.org/pdf/2301.10743.pdf).

---

> ### Author Response · Authors · 2023-11-19
>
> Q: Many related works are missing. The paper states that “there has been very little research on identifying logical languages that can be accepted by transformers”. However, with a quick google scholar search, I found the following highly-related papers not cited in the paper. It would be nice to discuss the relation of this paper’s results with these prior works.
> [1] Merrill, William, Ashish Sabharwal, and Noah A. Smith. "Saturated transformers are constant-depth threshold circuits." Transactions of the Association for Computational Linguistics 10 (2022): 843-856.
> [2] Merrill, William, and Ashish Sabharwal. "The parallelism tradeoff: Limitations of log-precision transformers." Transactions of the Association for Computational Linguistics 11 (2023): 531-545.
> [3] Strobl, Lena. "Average-Hard Attention Transformers are Constant-Depth Uniform Threshold Circuits." arXiv preprint arXiv:2308.03212 (2023).
>
> A: Here we were referring to "logical languages", i.e., languages that are defined by means of logical formulas (which do not always have natural circuit complexity classes counterparts, e.g., our FO(Mon) and LTL(+,C)). At the time of writing, the paper by Chiang et al. (2023) was the only example we were aware of, which we mentioned in Related Work. A new article on logical languages accepted by transformers appeared on arXiv last month (after our submission):
>
> Dana Angluin, David Chiang, Andy Yang. Masked Hard-Attention Transformers and Boolean RASP Recognize Exactly the Star-Free Languages. (https://arxiv.org/abs/2310.13897)
>
> An excellent survey also recently appeared on arXiv after the submission of our paper that discusses logical languages accepted by transformers:
>
> https://arxiv.org/abs/2311.00208
>
> We will discuss these in detail in the final version of our paper.
>
> None of [1]-[3] deal with logical languages, but rather connections of different classes of transformers to circuit complexity, like the earlier papers by Hahn (2020) and Hao et al. (2022), which we already discussed in detail in the introduction. [Note also that [3] appears after the submission of our paper, so we could not possibly cite it.] We have added [1]-[3] in the introduction.

---

### Official Review · Reviewer_wdZx · 2023-11-01

**Soundness:** 2 fair
**Presentation:** 2 fair
**Contribution:** 2 fair
**Rating:** 3
**Confidence:** 3

**Summary:**

This paper investigates whether circuit language can be accepted by existing transformer encoders with hard attention. Specifically, this work concludes that UHATs cannot accept all languages in $AC^0$, but they can still accept all languages in a ’monadic’ version. Besides,
this work finds out that AHATs, other transformer encoders, can express any language definable in a powerful counting logic. Moreover, this work provides sufficient theoretical justifications to support their findings and conclusions.

**Strengths:**

1. The study about whether circuit language can be accepted by existing transformer encoders is interesting and impressive.
2. The paper offers comprehensive theoretical justification to demonstrate and validate their findings.

**Weaknesses:**

1. The structure of this paper is very messy, which is very hard to follow. Let me take the Section Introduction as an instance:

1.a In Section 1 Introduction, the paper claims that "the expressive power of transformer encoders
has not been fully elucidated to date.". I am curious about that. What do you mean they are not fully elucidated?

1.b I am very confused about the challenges of studying the circuit language. I could not find any information to discuss the existing challenges and related works, which makes it hard to understand the motivation for this work.

1.c What are your contributions to this work? I could not find any conclusions about contributions after reading this section, or even the whole manuscript.


2. The paper has a very weak introduction to related works, making this work hard to understand and compare.

3. I would suggest that an illustration figure be provided to clearly show the main idea of this work.

**Questions:**

Please refer to the weakness of the questions that I proposed.

---

> ### Author Response · Authors · 2023-11-19
>
> Thanks for your questions. We provide responses below.
>
> Q: In Section 1 Introduction, the paper claims that "the expressive power of transformer encoders has not been fully elucidated to date.". I am curious about that. What do you mean they are not fully elucidated?
>
> A: We mean there is currently no precise characterization of which languages are accepted by different kinds of transformer encoders with infinite precision. We have rephrased this in the pdf to make it more understandable.
>
> Q: I am very confused about the challenges of studying the circuit language. I could not find any information to discuss the existing challenges and related works, which makes it hard to understand the motivation for this work.
>
> A: The question of what can and cannot be expressed by various classes of transformers lies at the heart of the foundation of the area. The works in the past five years (e.g. Hahn (2020) and Hao et al. (2022)) have underlined such an intimate connection between circuit complexity classes and transformer encoders, that results on circuit complexity (e.g. the inability of AC0 circuits to "count", i.e., solve PARITY) have been used to show an inherent limitation of various classes of transformer encoders, prominently Unique Hard Attention Transformers (UHAT) and Average Hard Attention Transformers (AHAT). Despite these, the precise expressivity of transformer encoders belongs to some of the most important open problems in the field, e.g., see Hao et al. (2022), as well as the following new excellent survey that has appeared on arXiv after the submission of our paper:
>
> https://arxiv.org/abs/2311.00208
>
> More precisely, any UHAT (resp. AHAT) transformer was known to be able to be simulated by AC0 (resp. TC0) circuits. Note that AC0 and TC0 are major complexity classes, and after decades of research the expressive power of these circuit classes is understood fairly well. This leads to the open problem of whether any AC0 (resp. TC0) language can also be recognized by UHAT (resp. AHAT). If not, point out which ones, as well as a natural subclasses of AC0 and TC0 languages that can be captured by UHAT and AHAT, respectively.
>
> Q: What are your contributions to this work? I could not find any conclusions about contributions after reading this section, or even the whole manuscript.
>
> A: In relation to our answer to the previous question, our results have significantly improved our understanding of what UHAT and AHAT transformer classes are (in)capable of. We have provided a concrete "counting language" in AC0 and showed that it cannot be recognized by UHAT, consequently UHAT is a strict subclass of AC0. We have also provided a natural subclass of AC0 that can be captured by UHAT, namely, a logical counterpart L of AC0 but restricted to monadic numerical predicates. We have shown similar results as well for AHAT by providing a natural subclass of TC0 (in terms of an extension of L that naturally incorporates counting capabilities of TC0, in particular the majority function) that can be captured by AHAT. As a corollary of these results, we showed that AHAT has an extremely powerful counting capability (e.g. the language of strings of a's, b's, and c's with the same number of occurrences of these letters).
>
> Q: The paper has a very weak introduction to related works, making this work hard to understand and compare.
>
> A: We feel that we have mentioned most of the papers that provide the relevant background for understanding our results, but of course we may have missed some (the area is, in fact, very dynamic). We will revise this for the final version of the paper. The fact that there is now a survey on the topic will definitely be useful for doing this.
>
> Q: I would suggest that an illustration figure be provided to clearly show the main idea of this work.
>
> A: Thanks. We will think about how to implement this idea in the final version of the paper.

---

> > ### Comment · Reviewer_wdZx · 2023-11-23
> > **Thank you for your clarification!**
> >
> > After going through your response and the revised manuscript, I would like to insist on my decision as a rejection. I believe this work still needs further revisions and the model should also be improved.

---

### Official Review · Reviewer_ceMz · 2023-11-08

**Soundness:** 3 good
**Presentation:** 2 fair
**Contribution:** 2 fair
**Rating:** 6
**Confidence:** 2

**Summary:**

Review
======

As a non-IA specialist but a circuit-complexity and automata specialist,
I found those connection amusing but slightly artificial.
I will not express myself on the pertinence of analizing the expressivity
of those models from IA-perspective but provides a bit of insight about
the automata/circuit complexity side.

First I found some claim dubious:

In a footnote you claim that only rational numbers where used in Hao (2022)
but generalize it real numbers.  It is known in complexity theory  that going from rational to real number is always complicated. Even
calculability theory can become weird when considering real numbers.

While it is rather clear that computation performed by UHAT can be computed
in AC0 when restricted to rational numbers, it is much less clear when going
within rational numbers. If it holds, it by the sake of some continuity and
approximation by real numbers arguments, but it deserves some details...
I believe it should be explained.

Because I don't really get that, I have assumed that the remaining where
done on rational numbers;

About the question about "what fragment of AC0, UHAT can belongs to", since it
contains all regular languages in AC0, it is improbable that you can find
a sound answer for that since it is already wide open for regular languages.
Indeed:
- regular language are complete for each level of the depth hierarchy AC0,
- regular language are in quasilinear AC0 and proving that they are linear AC0
is a long standing open problems (see the survey of Koucky on the topic).

About Proposition 1. I believe a much simpler argument can be used using
a simple padding argument:

>AC^0 recognized all languages up to exponential padding.
>That is, Pad(L) = { u #^{2^|u|} \mid u in L } is always in AC^0 whatever is L.
>Your UHAT shouldn't be able to capture all Pad(L) as it only can remember a small
>piece of information and convolute it. Basic information theory/pigeon hole might
>help to conclude without a hammer of sensitivy of circuits within AC0.

About proposition 5: I don't really understand why the Parikh
closure/permutation closure of languages is meaningful here. Sounds like an
arbitrary property to me.


Logic with counting operators has been introduced in the past (Majority logic)
and a study of its expressivity with respect to circuit classes.
See for instance (https://link.springer.com/chapter/10.1007/978-3-642-02737-6_7)
Is there a connection?

**Strengths:**

The paper contribute to a fun connection between formal language theory in order to analyze the expressivity of transformers. This line of research sounds more like research performed in TCS than in IA tracks but since it is apply to IA-defined model it makes some sense. Understanding the expressivity of those model might be enlightening to people actually playing with them.

**Weaknesses:**

Some of the contribution sounds really artificial to me. In particular, why should we study the commutative closure of language defined by transformers? While it makes sense in TCS, I fail to grasp the importance in this context.

**Questions:**

Comments
========
- The last sentence of the second paragraph of section 2.2 makes no sense.
f is a function from Sig -> R^d, T: Sig^+ -> R et the last sentence
says that T get an input sequence that type as a sequence of vectors of R"e.
The value of T(w) = (t, v0) only depending of t and v0 which is plain weird
as it gives the impression it depends of a constant (t) and v0 which is f(a_0) + p(0, n).
The whole paragraph is thus buggy.

- You can have some feelings on what is going on with logic extended with monadic
predicates through this paper.

https://dl.acm.org/doi/10.1145/3091124

- About your open question:
Additionally, does there exist a language in the circuit complexity class TC0, the
extension of AC0 with majority gates, that cannot be recognized by AHATs

I would go for Dyck language. Sounds hard for a AHATs.

---

> ### Author Response · Authors · 2023-11-19
>
> Many thanks for your valuable comments. They will definitely help in improving the paper. We respond to your questions next.
>
> Q: While it is rather clear that computation performed by UHAT can be computed in AC0 when restricted to rational numbers, it is much less clear when going within rational numbers ...
>
> A: Thank you for noticing, we should have made it more explicit. Actually, the paper by Hao et al. (2022) shows that even a more general model they call GUHAT, where arbitrary attention and activation functions over an arbitrary set of values are allowed, can be simulated efficiently in AC^0. This model subsumes UHAT with real numbers.
>
> Q: About Proposition 1. I believe a much simpler argument can be used using a simple padding argument: AC^0 recognized all languages up to exponential padding ...
>
> A: We actually considered this idea, but we do not see how to finish this argument. The problem is that actually information propagates substantially between vectors through the maximization of attention score. Another informal argument, why the situation is not that simple with this approach is that such an information-based argument should also apply to a more general model GUHAT considered in Hao et al. (2022). However, it is not hard to see that if the input contains only log n substantial variables, then we can compute the standard DNF within GUHAT: just compute all possible ANDs of literals on the first layer (one entry in the layer can compute one AND) and compute OR of them on the second layer. Of course, this does not work if we have a function with 2 log n substantial variables, since there are too many ANDs now. But it seems that a simple information theoretic argument should not be affected by the constant in front of the logarithm.
>
> Q: Logic with counting operators has been introduced in the past (Majority logic) and a study of its expressivity with respect to circuit classes. See for instance (https://link.springer.com/chapter/10.1007/978-3-642-02737-6_7) Is there a connection?
>
> A: It is difficult to try to establish a precise connection between our counting LTL and the numerous extensions of FO with counting that have been proposed in the literature. This is because very few extensions of Kamp’s theorem have been obtained for counting logics (the only one we are aware is given in the following paper: https://www.sciencedirect.com/science/article/pii/S1571066111001447?ref=pdf_download&fr=RR-2&rr=8268825c5e7f3687). In turn, our logic LTL + counting is subsumed by the infinitary counting logic presented in Chapter 8 of Libkin’s book “Elements of Finite Model Theory”. We do not know whether sensible connections can be established with less powerful logics, such as extensions of FO with majority.
>
> Q: Some of the contribution sounds really artificial to me. In particular, why should we study the commutative closure of language defined by transformers?
>
> A: Two themes have played an important role in understanding the expressive power of transformer encoders: (1) counting, and (2) regular languages. The result by Hao et al. (2022) shows that AHAT cannot recognize some regular language (under a reasonable complexity theoretic assumption). Likewise, Hahn's well-known result that the regular language PARITY is not in UHAT implies the inability of UHAT to "count" the number of 1s in the input string (i.e. eveness or oddness thereof).
>
> Results in our paper on Parikh images (Proposition 5) and permutation-closed languages (Corollary 2) provide a characterization of the expressive power of UHAT and AHAT in expressing counting properties, especially in relation to regular languages.
>
> Parikh-equivalence (i.e. equivalence up to letter permutation) is an established notion in automata theory, whereby a computing device (in our case, transformers) is allowed to try all letter permutations of the input string, and accepts iff some such letter-permutation is accepted. In other words, the "correct" reordering of the letters in the input word will have to be first nondeterministically guessed. Proposition 5 shows the ability of UHAT to capture regular languages (up to Parikh-equivalence), and particularly "counting regular languages" like PARITY can in this technical sense be expressed in UHAT.
>
> The ability to recognize permutation-closed languages is more powerful than Parikh-equivalence, in that the computing device is required to be able to perform the required counting task, *regardless* of the reordering of the letters in the  input string. Corollary 2 shows that AHAT can capture permutation closures of regular languages, and in this way capture languages of strings with the same number of a's, b's, and c's (which is by the way not even regular).

---

> > ### Comment · Reviewer_ceMz · 2023-11-22
> >
> > A: Thank you for noticing, we should have made it more explicit. Actually, the paper by Hao et al. (2022) shows that even a more general model they call GUHAT, where arbitrary attention and activation functions over an arbitrary set of values are allowed, can be simulated efficiently in AC^0. This model subsumes UHAT with real numbers.
> >
> > Ok, I didn't check it carefully. Somehow it probably shows that you can approximate your values by countable one without loosing on expressivity (which makes sense).  I will not look too carefully into those details, but it should be make precise in the paper, maybe?
> >
> > A:We actually considered this idea, but we do not see how to finish this argument. The problem is that actually information propagates substantially between vectors through the maximization of attention score.
> >
> > The information that propagate seems to still be of constant or log nature. It is actually hard to tell because I fail to understand the initial model (but this is my fault probably). Nevertheless, AC^0 can propagate information in "any direction", this is the essence of FO[Arb] result of Immerman. A small example: "there exists a,b,c in the word such that the index of a + index of b is the index of c". This example is kind-of annoying for sequential deterministic model. I would also be interested if you can can do log-bit counting like in AC^0 for instance. It seems also out of reach. (Those are not recommendation of update of your paper, but like, yardstick question about complexity within AC^0).
> >
> > A:Another informal argument, why the situation is not that simple with this approach is that such an information-based argument should also apply to a more general model GUHAT considered in Hao et al. (2022). However, it is not hard to see that if the input contains only log n substantial variables, then we can compute the standard DNF within GUHAT: just compute all possible ANDs of literals on the first layer (one entry in the layer can compute one AND) and compute OR of them on the second layer. Of course, this does not work if we have a function with 2 log n substantial variables, since there are too many ANDs now. But it seems that a simple information theoretic argument should not be affected by the constant in front of the logarithm.
> >
> > AC^0 is able to actually do poly-log. So it is not a multiplicative constant ... I don't really buy this argument.
> >
> > A: It is difficult to try to establish a precise connection between our counting LTL and the numerous extensions of FO with counting that have been proposed in the literature. This is because very few extensions of Kamp’s theorem have been obtained for counting logics (the only one we are aware is given in the following paper: https://www.sciencedirect.com/science/article/pii/S1571066111001447?ref=pdf_download&fr=RR-2&rr=8268825c5e7f3687). In turn, our logic LTL + counting is subsumed by the infinitary counting logic presented in Chapter 8 of Libkin’s book “Elements of Finite Model Theory”. We do not know whether sensible connections can be established with less powerful logics, such as extensions of FO with majority.
> >
> > Indeed, the fields is slightly a mess. LTL + Counting can be found in trees I believe (CTL*) but the context is vastly different.
> >
> > About last comment: I am well aware about the importance of Permutation close regular language in automata theory. But while the general expressiveness of your model is interesting, those results are IMO a bit ad-hoc and incremental.

---

> ### Author Response · Authors · 2023-11-19
>
> Q: The last sentence of the second paragraph of section 2.2 makes no sense. f is a function from Sig -> R^d, T: Sig^+ -> R et the last sentence says that T get an input sequence that type as a sequence of vectors of R"e ...
>
> A: In our view, the main point of confusion arises from the fact that we are abusing notation and using T to denote both the transformer, which transforms sequences of vectors in R^d to sequences of vectors in R^e, and the function that given a word \bar w outputs a real number based on T. We have modified this and updated the pdf accordingly. Regarding your comment that the value of T(w) only depends on t and v0: please notice that v0 is not f(a_0) + p(0, n), but the first vector in the output of T when presented with the input f(a_0) + p(0, n), …, f(a_n) + p(n,n). We believe there is no mistake in this case.
>
> Q: About your open question: Additionally, does there exist a language in the circuit complexity class TC0, the extension of AC0 with majority gates, that cannot be recognized by AHATs: I would go for Dyck language. Sounds hard for a AHATs.
>
> A: Thanks for the suggestion! In fact, [Yao et al. 2021] show that Dyck languages (even with more than one type of brackets) can be accepted by a transformer with soft attention. But they are actually using only uniform weights, which can also be done in AHAT. That being said, they are also using layer normalization. It seems unknown whether Dyck with more than one type of brackets can be done by AHATs without layer normalization.

---

### Meta-Review · Area_Chair_gB9d · 2023-12-05

**Metareview:**

The paper contributes to the understanding of the expressiveness of transformer encoders with attention. From the point of view of formal languages, it brings limited significant results. However, it bridges communities and carries interesting conclusions for the machine learning point of view, and can raise new results that contribute to better characterize the expressivity of classes of neural networks.

The main contributions concern Unique Hard Attention Transformers (UHATs) and their relation to Boolean circuits. They prove that UHATs cannot accept all languages in AC^0, but the main result shows that UHATs can recognize all languages definable in first-order logic (FO) with unary numerical predicates only FO(Mon). Then AHATs (average head attention transformers) capture all regular languages and have some counting ability. They are shown to be as expressive as linear temporal logic with unary numerical predicates equipped with counting features.

**Justification For Why Not Higher Score:**

The tools used in this paper are not well known to many ICLR participants. While the paper may contribute to their discovery and establish stronger links with the field of language theory and circuit complexity, it runs the risk of not finding its audience at ICLR.

**Justification For Why Not Lower Score:**

The paper has interesting conclusions from a machine learning point of view, and can provide new results that help to better characterize the expressiveness of the attention mechanism.

---

### Decision · Program_Chairs · 2024-01-16

Accept (poster)